# Hygroscopic growth characteristics of anthropogenic aerosols over central China revealed by lidar observations

Dongzhe Jing [1,2,3], Yun He[1,2,3,*], Zhenping Yin[4], Detlef Müller[4], Kaiming Huang[1,2,3], Fan Yi[1,2,3]

[1]School of Earth and Space Science and Technology, Wuhan University, Wuhan 430072, China

[2]Key Laboratory of Geospace Environment and Geodesy, Ministry of Education, Wuhan 430072, China.

[3]State Observatory for Atmospheric Remote Sensing, Wuhan 430072, China.

[4]School of Remote Sensing and Information Engineering, Wuhan University, Wuhan 430072, China

*Correspondence to*: Yun He (heyun@whu.edu.cn)

**Abstract.** Lidar-derived particle backscatter coefficient is commonly used to assess air pollution levels; however, hygroscopic
growth can amplify particle backscatter and hinder accurate assessment of particle concentration. This study investigated the hygroscopic growth characteristics of urban anthropogenic aerosols in Wuhan (30.5°N, 114.4°E), central China, using ground-based 532-nm polarization lidar observations during 2010-2024. A total of 192 cases were identified based on the following criteria: (1) the presence of a layer thicker than 300 m; (2) a lidar-derived backscatter coefficient that increases monotonically with simultaneously-measured relative humidity (RH) from radiosonde, and (3) limited variations in key meteorological
parameters, including water vapor mixing ratio, potential temperature, and wind speed and direction. Using the Hänel parameterization method, the hygroscopic growth parameter $\gamma$ was estimated as 0.62 ($\pm$0.24), corresponding to a backscatter coefficient enhancement factor of 2.36 at 85% RH. No evident differences in $\gamma$ were observed between the boundary layer (0.63$\pm$0.25) and free troposphere (0.60$\pm$0.24). The annual mean $\gamma$ increased from 0.49 in 2014 to 0.63 in 2017 and stabilized within 0.6-0.7 after 2018, closely following the evolution of the annual mean $NO_2$-to-$SO_2$ concentration ratio. The minimum
seasonal average $\gamma$ occurred in winter (0.56), while the maximum was observed in autumn (0.64). These results provide a comprehensive characterization of the long-term and seasonal hygroscopicity of pollutants over central China, enhancing our understanding of the influence of hygroscopic growth on lidar-observed particle backscatter coefficients and offering valuable insights for urban air pollution control strategies.

 **1 Introduction**

Atmospheric aerosols impact global climate directly by scattering or absorbing solar radiation (Liu and Matsui, 2021), and indirectly via aerosol-cloud interactions by acting as cloud condensation nuclei (CCN) or ice-nucleating particles (INP) (Rosenfeld et al., 2014; He et al., 2021, 2022). In the atmosphere, soluble aerosols can take up water vapor under high relative humidity (RH) conditions, causing them to grow in size through so-called hygroscopic growth (Hänel, 1976). This process alters aerosols' optical and microphysical properties, thus changing their impact on climate. Ji et al. (2025) found that the aerosol infrared radiation effect in the Arctic is 1.45 W·m$^{-2}$ under dry atmospheric conditions, which increases 7-fold when RH is between 60-80% and even up to 20 times when RH exceeds 80%. In addition, aerosol hygroscopicity plays a vital role in activating cloud droplets, with activation efficiencies of 0-34% for low hygroscopicity particles and 57–83% for high hygroscopicity particles, respectively (Väisänen et al., 2016). Furthermore, in urban environments, when aerosols take up water vapor and grow, the atmospheric visibility could significantly reduce, leading to severe haze events (Liu et al., 2013; Chen et al., 2019). Aerosol hygroscopicity also affects the deposition efficiency in the respiratory tract and influences human health (Sorooshian et al., 2012).

The hygroscopic enhancement factor $f(\mathrm{RH})$ quantitatively describes aerosol hygroscopicity, defined as the ratio of the particle scatter or backscatter coefficient (Granados-Muñoz et al., 2015; Zhang et al., 2024) or particle diameter (Zieger et al., 2013) at a given RH to the corresponding values under dry conditions (typically ~40%) (Hänel, 1976; Titos et al., 2016). To determine this factor, in-situ measurements commonly use humidified nephelometers (Zieger et al., 2010, 2013) and humidified tandem differential mobility analyzer (HTDMA) (Deshmukh et al., 2025; Yu et al., 2025). These instruments first dry the collected aerosol samples and then re-humidify them to a target RH, enabling comparisons of scatter coefficients or particle diameters under dry versus humid conditions. This approach provides general insights into the hygroscopicity of different aerosol types. However, this process can be affected by particle sampling losses (Titos et al., 2016) and by variations in aerosol size or scatter coefficient (e.g., deliquescent aerosols, Zieger et al., 2016). In addition, accurately determining hygroscopic enhancement factors at RH above 90% remains challenging for both nephelometers and HTDMA (Lv et al., 2017).

In addition to in situ measurements, lidar also has the capability to measure aerosol hygroscopicity, a technique first demonstrated by Ferrare et al. (1998). From then on, lidar-based experiments have been widely used to derive the hygroscopic particle backscatter coefficient enhancement factor $f_\beta(\mathrm{RH})$, (Granados-Muñoz et al., 2015; Haarig et al., 2017, 2025; Sicard et al., 2022; Miri et al., 2024; Veselovskii et al., 2025; Zhang et al., 2025), defined as the ratio of the particle backscatter coefficient at a given RH to that under dry conditions (e.g., 40%). Unlike in situ measurements, lidar observations provide hight-resolved particle backscatter coefficient with high vertical resolution, and thus, enable the estimation of aerosol hygroscopicity under real atmospheric environments and within the entire atmospheric column, instead of in controlled laboratory settings. However, lidar-based studies of aerosol hygroscopic growth require simultaneous profiles of meteorological parameters as input (Granados-Muñoz et al., 2015); as a result, most existing lidar studies have focused on individual case analyses rather than long-term monitoring. At our observatory in Wuhan (30.5° N, 114.4° E), a megacity in

central China, continuous (24/7 and regardless of bad weather conditions), long-term (except for the time of hardware maintenance) lidar observations have been conducted since 2010 (Yin et al., 2021; He et al., 2024; Jing et al., 2024, 2025). This dataset, spanning more than a decade, provides a solid basis for a statistical analysis of the hygroscopicity growth characteristic of urban anthropogenic aerosols.

Due to the rapid urbanization and industrialization in China since the early 21st century, high aerosol loading and complex pollutants have attracted increasing attention (Xie et al., 2016). In response, the Chinese government has implemented a series of emission control policies. Our earlier study showed that annual variations in the anthropogenic aerosol optical depth (AOD) at 532 nm over Wuhan during the past 15 years can be divided into two stages: a rapid decline with a rate of -0.068 $yr^{-1}$ from 2010 to 2017, followed by a fluctuation period from 2018 to 2024 (Jing et al., 2025). These findings highlight that emission control policies were highly effective in the first stage, whereas their impact weakened in the second stage. We also identified an imbalance in $SO_2$ and $NO_2$ emission reductions, with the $NO_2$-to-$SO_2$ concentration ratio rising sharply from 1.8 in 2014 to 5.3 in 2017 (Jing et al., 2025). This shift likely promoted the formation of secondary aerosols (e.g., particulate nitrate) and partly explained the cessation of the declining trend in the second stage (Liu et al., 2018). Therefore, it is essential to investigate how aerosol hygroscopicity responds to changes in pollutant components, for interpreting the long-term AOD patterns.

In this study, we statistically analyze the hygroscopic growth characteristics of anthropogenic aerosols from 2010 to 2024 using ground-based polarization lidar observations over Wuhan, together with associated radiosonde and reanalysis meteorological data. This paper is organized as follows. Section 2 briefly describes the adopted instruments and data processing methods. Section 3 presents a case study illustrating the identification of aerosol hygroscopic growth cases and the estimation of the hygroscopic growth parameter. Section 4 offers a statistical analysis of the hygroscopic growth parameter $\gamma$ for anthropogenic aerosols over Wuhan. The last section summarizes the main findings and presents the conclusions.

## 2 Instrumentation and data

### 2.1 Study area

Wuhan (30.5° N, 114.4° E) is a major industrial city in central China with a population of over 13 million, producing abundant urban anthropogenic emissions from sources such as vehicle exhaust and industrial fossil fuel combustion (Zhang et al., 2015a). The city lies in the subtropical monsoon region and experiences four distinctive seasons. In winter, the northeast monsoon brings cold, relatively dry weather (Wu and Wang, 2002). Frequent temperature inversions in the lower troposphere suppress vertical convection and often lead to severe air pollution (Zhang et al., 2021). In summer, the southeast and southwest monsoons cause high temperatures and heavy rainfall, creating favorable conditions for pollution dispersion (Ding et al., 2015). Spring and autumn are the transition phases between these two regimes. Additionally, regional air mass transport also plays a significant role in aerosol loading over Wuhan. In spring and winter, mineral dust from deserts in north and northwest China as well as in Mongolia is frequently long-range transported to Wuhan (He and Yi, 2015; Jing et al., 2024). In summer and

autumn, agricultural biomass burning in neighboring provinces sometimes contributes to the poor air quality and severe haze
events over Wuhan (Zhang et al., 2014; Jing et al., 2025).

Among the major aerosol types in Wuhan, mineral dust is generally considered hydrophobic. In contrast, all the non-dust components are classified as anthropogenic aerosols, which exhibit varying degrees of hygroscopicity due to the inclusion of water-soluble inorganic ions and organic matter. Our lidar site is located in central Wuhan, an area surrounded by an extensive water network (the Yangtze River and numerous urban lakes; Figure 1), which creates a humid atmospheric environment that
facilitates aerosol water uptake. Therefore, particle hygroscopic growth may contribute to the lidar-derived particle backscatter coefficients. In this study, all lidar-derived aerosol optical properties discussed, including the particle backscatter coefficient, extinction coefficient, and AOD, are attributed exclusively to anthropogenic aerosols.

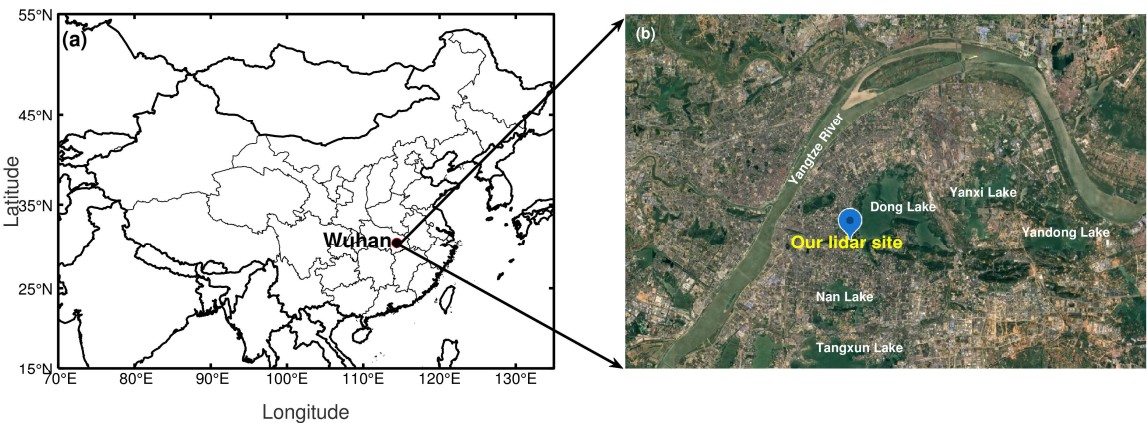

**Figure 1. Location of Wuhan and our lidar site (Google Earth©, 2025).**

**2.2 Polarization lidar and data processing**

Height-resolved aerosol optical properties over Wuhan have been observed using a 532-nm ground-based polarization lidar since October 2010 (Yin et al., 2021; He et al., 2024; Jing et al., 2024, 2025). Detailed specifications of the lidar system were provided in Kong and Yi (2015). In 2017, a transparent waterproof window was installed on top of the lidar container, enabling uninterrupted lidar operations regardless of rainy or snowy weather conditions (Yi et al., 2021). Raw data are stored with a
time resolution of 1 minute and a vertical resolution of 30 meters. The lowermost height with complete field-of-view (FOV) observation is 0.3 km. Specifications of the polarization lidar system are listed in Table 1.

**Table 1. Specifications of the polarization lidar system at Wuhan University. (He et al., 2024).**

| Transmitter | | Receiver | |
|---|---|---|---|
| Laser | Continuum Inlite II-20 | Telescope | 300 mm Cassegrain |
| Wavelength | 532 nm | Diameter | 300 mm |
| Energy/pulse | ~ 120 mJ | Field of view | 1 mrad |

| Repetition rate | 20 Hz | PMT | Hamamatsu 5783P |
|---|---|---|---|
| Pulse duration | 6 ns | Digitizer | Licel TR40-160 |

The volume depolarization ratio $\delta_v$ (VDR) is defined as the ratio of perpendicular- to parallel-oriented signals, multiplied by the gain ratio between the two polarized channels, and is used to derive the particle depolarization ratio $\delta_p$ (PDR) (Freudenthaler et al., 2009). The particle backscatter coefficient $\beta_p$ and particle extinction coefficient $\alpha_p$ are retrieved using the Fernald method (Fernald, 1984), assuming a fixed lidar ratio of 50 sr (Wang et al., 2016). In addition, the non-dust particle backscatter coefficient $\beta_{nd}$ is calculated using the polarization-lidar photometer networking (POLIPHON) method as follows (Tesche et al., 2009; Mamouri and Ansmann, 2014):

$$\beta_{nd}(z) = \beta_p(z) - \beta_p(z)\frac{[\delta_p(z) - \delta_{nd}](1 + \delta_d)}{(\delta_d - \delta_{nd})[1 + \delta_p(z)]} \tag{1}$$

where z represents the altitude; $\delta_d = 0.31$ and $\delta_{nd} = 0.05$ are the particle depolarization ratios for dust and non-dust, respectively. For each profile, we set $\beta_{nd}(z) = \beta_p(z)$ if $\delta_p(z) < \delta_{nd}$ and $\beta_{nd}(z) = 0$ if $\delta_p(z) > \delta_d$ (Mamouri and Ansmann, 2014). As noted in the previous section, the non-dust component primarily corresponds to anthropogenic aerosols over Wuhan; thus, $\beta_{nd}$ and $\alpha_{nd}$ represent the optical properties of anthropogenic aerosols. Table 2 lists the uncertainties in the lidar-derived aerosol optical property parameters.

**Table 2. Estimated uncertainties of the lidar-derived optical properties at 532 nm (Jing et al., 2025).**

| Parameter | Uncertainty |
|---|---|
| Volume depolarization ratio $\delta_v$ | <5% |
| Particle depolarization ratio $\delta_p$ | 5-10% |
| Particle backscatter coefficient $\beta_p$ | <10% |
| Particle extinction coefficient $\alpha_p$ | <20% |
| Non-dust backscatter coefficient $\beta_{nd}$ | 10-30% |
| Non-dust extinction coefficient $\alpha_{nd}$ | 30-40% |

Cloud-free profiles with signal accumulation times of 30-80 minutes are obtained using a cloud screening algorithm (Yin et al., 2021). For each cloud-free profile, the particle backscatter coefficient (i.e., total, dust, and non-dust components), and both volume and particle depolarization ratios are retrieved. We utilize the same methodology as Yin et al. (2021) and extend the aerosol profiles to September 2024 (Jing et al., 2025). A total of 24910 cloud-free profiles are identified from 2139 observational days between October 2010 and September 2024. To avoid potential contamination from severe haze or fog, cloud-free profiles containing one or more vertical bins with extinction coefficients exceeding 1.5 km$^{-1}$ are excluded. In total, 676 (2.7% of all) cloud-free profiles are removed from the analysis. This mature dataset is used to further investigate hygroscopic growth.

**2.3 HYSPLIT model**

The Hybrid Single Particle Lagrangian Integrated Trajectory (HYSPLIT), developed by the National Oceanic and Atmospheric Administration Air Resources Laboratory (NOAA ARL), was used to simulate both forward and backward air mass trajectories (HYSPLIT, 2025). These simulations are driven by meteorological field data from the GDAS archive (Kanamitsu, 1989) and require initialization parameters such as start time, altitude, and geographical location (Draxler and Rolph, 2003; Stein et al., 2015). In this study, three backward trajectories arriving at Wuhan at different altitudes were

simulated to trace the potential origins of aerosols at those altitudes.

**2.4 Radiosonde data**

    Two radiosonde launches were conducted daily at 0800 local time (LT) and 2000 LT, at 30.6° N, 114.1° E, approximately 24 km away from our lidar site. The sondes measured vertical profiles of temperature, pressure, relative humidity (RH), water vapor mixing ratio, and wind speed/direction from the surface to up to ~30 km altitude. The measurement error for temperature

is less than 1 °C, and the uncertainty in RH is below 5% when the temperature exceeds -10 °C (Nash et al., 2011). The potential temperature θ is defined as (Bolton, 1980):

$$\theta = T \cdot \left(\frac{P_0}{P}\right)^{0.286} \tag{2}$$

where T is the temperature (K), P is the atmospheric pressure (hPa), and $P_0$ is the reference pressure of 1000 hPa. In this study, radiosonde data were interpolated to match the corresponding altitude bins of the lidar profiles using a cubic spline

interpolation method. This interpolation ensured consistent alignment between radiosonde and lidar measurements, facilitating our analysis of aerosol hygroscopic growth.

**2.5 ERA5 reanalysis data**

    The European Center for Medium-Range Weather Forecasts (ECMWF) reanalysis version 5 (ERA5) (Copernicus Climate Change Service, 2025) provides global atmospheric reanalysis data from January 1940 onward. ERA5 combines model outputs

with worldwide observations into a globally consistent and physically constrained dataset (Hersbach et al., 2020). It offers hourly estimates of atmospheric, land, and oceanic climate variables. The boundary layer height (BLH) is the depth of the boundary layer (BL) directly affected by dynamic, thermal, and other surface interactions (Peng et al., 2023). Above the BL is the free troposphere (FT), where aerosols primarily originate from non-local sources (Bourgeois et al., 2018). In this study, ERA5 hourly BLH data (Hersbach et al., 2023) for Wuhan were used to distinguish respective hygroscopic growth cases

occurring in the BL and FT. Considering the presence of an aerosol residual layer, the diurnal maximum BLH from ERA5 was adopted as the boundary between the BL and FT.

**3 Methodology of estimating the hygroscopic growth parameter**

Veselovskii et al. (2009) established a methodology to quantitatively estimate aerosol hygroscopicity from lidar measurements under conditions of increasing particle backscatter coefficient with altitude and a constant water vapor mixing ratio, which has been applied and refined in subsequent studies (Granados-Muñoz et al., 2015; Navas-Guzmán et al., 2019; Sicard et al., 2022). In this study, dust and non-dust (anthropogenic) aerosols are assumed to be externally mixed, with their optical properties considered relatively independent. Accordingly, the variation of the anthropogenic particle backscatter coefficient $\beta_{nd}$ with RH can be analyzed separately. The particle backscatter coefficient enhancement factor $f_\beta$ (RH) is defined as the ratio of particle backscatter coefficient at a given RH to that under dry conditions (Hänel, 1976):

$$f_\beta(\text{RH}) = \frac{\beta_{nd}(\text{RH})}{\beta_{nd}(\text{RH}_{dry})} \tag{3}$$

Lidar-derived cloud-free profiles within 2 hours before or after the radiosonde launches (around 08 or 20 LT) were selected (Sicard et al., 2022). To estimate $f_\beta$(RH), we identified the aerosol layers exceeding 300-m thickness and fulfilling the criterion that $\beta_{nd}$ increase monotonically with simultaneously measured radiosonde RH within the layer. The analysis was limited to altitudes below 7 km. The minimum $\beta_{nd}$ within an identified aerosol layer was required to exceed 0.5 Mm$^{-1}$sr$^{-1}$ to reduce interference from low signal-to-noise ratios. In addition, the maximum variations in radiosonde meteorological parameters within the identified aerosol layer were constrained as follows to ensure analysis under well-mixed atmospheric conditions (Sicard et al., 2022):

(1) △water vapor mixing ratio (WVMR) < 2 g kg$^{-1}$;

(2) △potential temperature (θ) < 2K;

(3) △wind speed (WS) < 2 m s$^{-1}$;

(4) △wind direction (WD) <15°.

These criteria ensure that the observed increase in $\beta_{nd}$ was solely due to particle growth through hygroscopic water uptake, rather than additional emissions or changes in aerosol composition (Granados-Muñoz et al., 2015).

Figure 2 shows an aerosol hygroscopic growth case observed at 1830-1900 LT on 19 July 2019. As altitude increased from 0.4 km to 1.2 km, $\beta_{nd}$ rose from 2.4 Mm$^{-1}$sr$^{-1}$ to 3.6 Mm$^{-1}$sr$^{-1}$, while RH increased from 54% to 82%. In contrast, the PDR gradually decreased from 0.07 to 0.04, indicating that the particles became more spherical due to water uptake (Miri et al., 2024). The maximum variations in WVMR, θ, WS, and WD were 0.64 g kg$^{-1}$, 0.28 K, 1.06 m s$^{-1}$, and 9.49°, respectively, reflecting a homogeneous aerosol layer under well-mixed atmospheric conditions. Three two-day backward trajectories, initialized at 1900 LT on 19 July at altitudes of 0.5, 0.8, and 1.2 km, all traced back to the coastal region northeast of Wuhan (Figure 3), suggesting a similar aerosol source across these altitudes. This case is therefore considered representative of the hygroscopic growth behavior of anthropogenic aerosols over Wuhan.

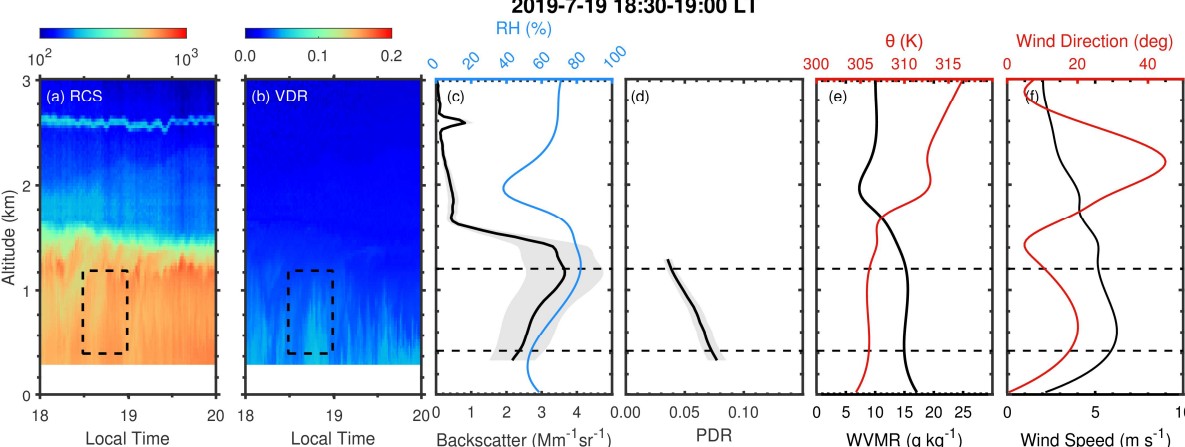

**Figure 2.** Time-height contour plots of (a) range-corrected signal (RCS) and (b) volume depolarization ratio (VDR) measured by polarization lidar over Wuhan at 18:00-20:00 local time (LT) on 19 July 2019. Profiles of (c) backscatter coefficient and relative humidity (RH), (d) particle depolarization ratio (PDR), (e) water vapor mixing ratio (WVMR) and potential temperature, (f) wind speed and direction. The lidar-derived profiles in (c) and (d) are obtained during 1830-1900 LT. All the meteorological parameter profiles are obtained from the radiosonde launched at around 2000 LT on the same day. The grey-shaded areas around the lidar profiles in c) and d) denote the uncertainty of lidar-derived optical properties.

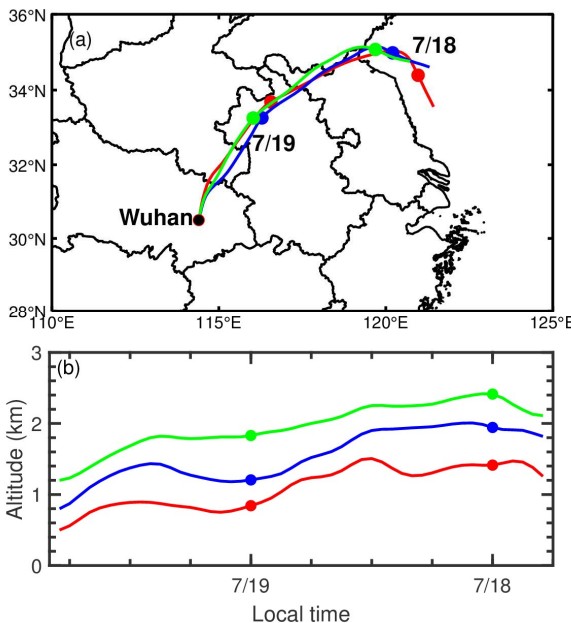

**Figure 3.** HYSPLIT three two-day backward trajectories starting from Wuhan (30.5° N, 114.4° E) at 1900 LT on 19 July 2019 at altitudes of 0.5, 0.8, and 1.2 km. The solid dots represent 00 LT for each day.

For the selected case, the particle backscatter coefficient enhancement factors $f_\beta$ (RH) for each altitude bin were calculated using Eq. (3), with a minimum RH ($RH_{min}$) of 54%, as shown by the black dots in Figure 4. To obtain $f_\beta$ (RH) for any RH > $RH_{min}$, parameterization of the relationship between $f_\beta$ (RH) and RH is required. Titos et al. (2016) evaluated 11 parameterization fitting methods based on nephelometer measurements and found that, for ambient aerosols, the differences among fitting curves were small, with most showing good agreement with measurements. Among them, the Hänel parameterization has been proved to be feasible and is now widely applied to estimate aerosol hygroscopic growth using lidar and radiosonde data (Veselovskii et al. 2009; Granados-Muñoz et al. 2015; Pérez-Ramirez et al., 2021; Sicard et al., 2022). To ensure comparability, we adopted the Hänel parameterization to fit $f_{\beta-Hänel}$(RH) starting from $RH_{min}$ as given by (Hänel, 1976):

$$f_{\beta-Hänel}(RH) = \left( \frac{1 - \frac{RH}{100}}{1 - \frac{RH_{min}}{100}} \right)^{-\gamma} \tag{4}$$

where $\gamma$ is the hygroscopic growth parameter that characterizes aerosol hygroscopicity. In this case, the fitting yielded an R-square of 0.99, indicating excellent agreement with the observations (black line in Figure 4). The derived $\gamma$ value of 0.48 suggests that the particles are moderately hygroscopic, typical of urban pollution (Bedoya-Velázquez et al., 2018). This result further indicates that although the trajectory traced back to coastal regions, marine aerosols (e.g. sea salt) have been largely removed by sedimentation, leaving an extremely limited influence of sea salt in Wuhan.

Although $\gamma$ is independent of $RH_{min}$, $f_{\beta-Hänel}$(RH) depends on the specific $RH_{min}$ chosen in each case. To ensure comparability and consistency, we define $f_{ref}$(RH) as the particle backscatter coefficient enhancement factor referenced to a unified RH value ($RH_{ref}$). Both $f_{ref}$(RH) and its parameterized fitting $f_{ref-Hänel}$(RH) can be extrapolated from $f_\beta$(RH) and $f_{\beta-Hänel}$(RH) using the following equations (Sicard et al., 2022):

$$f_{ref}(RH) = f_\beta(RH) \left( \frac{1 - \frac{RH_{min}}{100}}{1 - \frac{RH_{ref}}{100}} \right)^{-\gamma} \tag{5}$$

$$f_{ref-Hänel}(RH) = \left( \frac{1 - \frac{RH}{100}}{1 - \frac{RH_{ref}}{100}} \right)^{-\gamma} = f_{\beta-Hänel}(RH) \left( \frac{1 - \frac{RH_{min}}{100}}{1 - \frac{RH_{ref}}{100}} \right)^{-\gamma} \tag{6}$$

where $RH_{ref}$=40%. For RH < 40%, $f_{ref}$(RH) was set to 1. This assumption may underestimate $f_{ref}$(RH) by up to 25% for highly hygroscopic aerosols ($\gamma$>1), and by 10-15% for moderately hygroscopic aerosols ($\gamma$<0.5) (Titos et al., 2016). Moreover, the uncertainty in the particle backscatter coefficient enhancement factor can reach 38% at RH>95% (Adam et al., 2012). The resulting $f_{ref-Hänel}$(RH) is shown in Figure 4 by the blue dashed curve. At RH=85%, $f_{ref-Hänel}$(RH) is 1.93, indicating that the backscatter coefficient increases by a factor of 1.93 compared to dry conditions as RH increases from 40% to 85%.

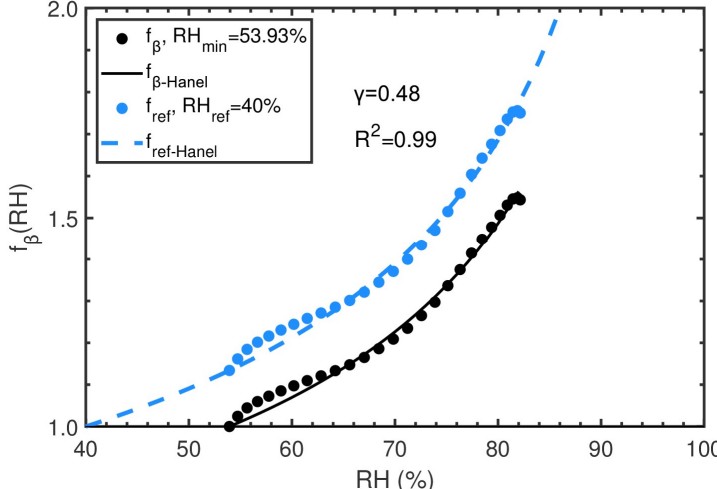

**Figure 4.** **The particle backscatter coefficient enhancement factors between RH values of 54% and 82% (black dots) and the**
**corresponding Hänel fit (black line). The extrapolated particle backscatter coefficient enhancement factors and Hänel fit referenced**
**to $RH_{ref}$=40% are also shown (blue dots and dashed line).**

In this study, only anthropogenic aerosols were considered, while the influence of natural aerosols, such as mineral dust or sea salt, was excluded. Hygroscopic growth parameters for mineral dust are known to be very low, with $\gamma$ values of 0.20 at 355 nm and 0.12 at 1064 nm (Navas-Guzmán et al., 2019). For marine aerosols, $\gamma$ has been estimated as 1.49 for pure sea salt (Haarig et al., 2017) and 1.1 for a mixture of sulfate and sea salt (Granados-Muñoz et al., 2015). As Wuhan is an inland city far from the ocean, the impact of marine aerosols is minimal; therefore, cases with $\gamma$ <0.2 or $\gamma$ >1.1 were treated as outliers and excluded from the analysis.

## 4 Results

### 4.1 Statistics of hygroscopic growth parameter

Figure 5 presents the probability density distribution of the particle backscatter coefficient enhancement factor $f_{ref}$(RH) for 192 identified cases from lidar observations during 2010-2024. The average $R^2$ value of 0.91 indicates a good fit with the Hänel parameterization for most cases. The average $\gamma$ value was 0.62±0.24, represented by the blue curve and shaded area, corresponding to $f_{ref-Hänel}$(85%) of 2.36, with a range of 1.69-3.29 when including the standard deviation. The $\gamma$ value in the BL (0.63±0.25) was comparable to that in the FT (0.60±0.24). The mean RH in the lower and middle troposphere was 71.2±12.5% (68.4±12.0% in the BL and 75.3±12.1% in the FT), corresponding to the highest probability density area (in yellow). This suggests that most hygroscopic growth of urban anthropogenic aerosols in Wuhan occurred under high RH conditions of around 60-80%. The particle backscatter coefficient enhancement factor $f_{ref-Hänel}$(85%) exhibited a broad distribution from 1.32 to 4.53, corresponding to $\gamma$ values of 0.20-1.09, likely reflecting the diverse hygroscopicity properties

of urban particles in Wuhan. Rapid urbanization over the past decades has exposed the city to numerous pollutants, including
245 various water-soluble inorganic ions, elemental carbon, and organic matter etc (Zhang et al., 2015a). Both chemical composition and particle size significantly influence aerosol hygroscopicity characteristics (Zieger et al., 2010, 2013), resulting in a variety of hygroscopic aerosol types.

In the previous section, dust and anthropogenic aerosols are assumed to be externally mixed. However, the internally mixing conditions are unavoidable for East Asian dust events (Xu et al., 2020), which may lead to misclassification of "coated dust"
as "anthropogenic" by the POLIPHON method. To assess the potential influence of internally mixed dust, a sensitivity analysis was conducted. Given that the $\delta_p$ during dust events over Wuhan varies between 0.1 and 0.3 (Jing et al., 2024), the threshold value for pure dust $\delta_d$ (in Eq. (1)) was reduced to lower values of 0.10~0.25, thereby making the extraction of "anthropogenic aerosols" more conservative. As a result, the $\gamma$ increases only slightly by 0.02 (0.62 to 0.64). Moreover, approximately 54.7% cases in this study show no dust interference, and for the remaining cases the dust optical depth (DOD) does not exceed 0.05.
It can be concluded that the error introduced by potential misclassification is limited.

The annual mean $\gamma$ from 2010-2024 is presented in Figure 5d, with the number of identified cases for each year indicated at the top of each bar. The annual mean $\gamma$ generally ranged from 0.5 to 0.7. Notably, the annual mean $\gamma$ sharply increased from 0.49 in 2015 to 0.63 in 2017, and stabilized high between 0.6 and 0.7 after 2018. The evolution of the annual mean $NO_2$-to-$SO_2$ concentration ratio in Wuhan from 2014 to 2024 is also presented in figure 5d (red dashed broken line), which closely
follows the trend of $\gamma$. The $NO_2$-to-$SO_2$ concentration ratio increased sharply from 1.8 in 2014 to 5.3 in 2017, and varied between 4 and 6 during 2018-2024, suggesting that the disparity in emission control measures for these two gaseous precursors, i.e., favoring more particulate nitrate formation, likely contribute to the increase in $\gamma$ after 2017. Chen et al. (2019) reported that a higher nitrate fraction in an aerosol mixture enhances aerosol hygroscopicity under the same RH conditions. In addition, our previous study showed that the anthropogenic AOD at 532 nm over Wuhan declined during 2010-2017 due to strict
emissions control policies, but this downward trend ceased from 2018 onwards (Jing et al., 2025). The relatively higher $\gamma$ values post-2017 result in a larger backscatter coefficient enhancement factor, indicating that AOD values after 2017 contain more contributions from hygroscopic growth effect, which partly offsets the effectiveness of emission control policies.

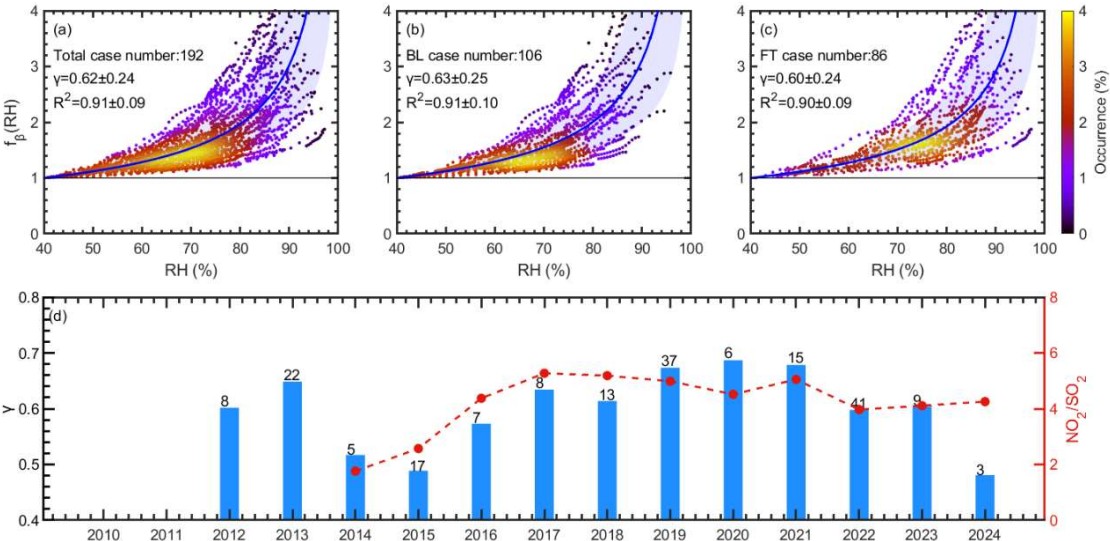

**Figure 5. Probability density distribution of the particle backscatter coefficient enhancement factors ($RH_{ref}$=40%) of (a) 192 selected hygroscopic growth cases describing the aerosol conditions in the low and middle troposphere (0-7 km); (b) 106 cases in the boundary layer; (c) 86 cases in the free troposphere during 2010-2024. The Hänel fits (blue solid line) were calculated with the mean hygroscopic growth parameter $\gamma$ =0.62. The shaded area represents the standard deviation of the Hänel fit line. (d) Bar plot of the annual mean $\gamma$ during 2010-2024. The case number for each year is presented at the top of each bar. The red dashed line represents the evolution of the annual mean $NO_2$-to-$SO_2$ concentration ratio during 2014-2024 (Jing et al., 2025).**

Figure 6 presents the seasonal variation of the particle backscatter coefficient enhancement factors, derived from the 192 identified cases. Seasons are defined as spring (March-April-May), summer (June-July-August), autumn (September-October-November), and winter (December-January-February). Most cases occurred in summer (80) and autumn (81), as dust is commonly present in spring and winter (Jing et al., 2024) and was therefore excluded from the analysis. There is no significant seasonal difference in $\gamma$, with a maximum of 0.64 in autumn and a minimum of 0.56 in winter, corresponding to $f_{ref-Hänel}(85\%)$ values of 2.43 and 2.17, respectively, i.e., a difference of approximately 11%. Urban pollutants emitted from human activities do not vary substantially with seasons, except for nitrate, which shows the highest fraction in $PM_{2.5}$ during winter (Zhang et al., 2015a).

To explain why the fraction of nitrate in Wuhan is the highest in winter, while $\gamma$ is the lowest (0.56), the mean layer heights (MLH) for cases in the four seasons are presented in figure 6. The MLH in winter is 2.5 km, higher than the 1.4-1.5 km observed in other seasons, likely due to frequent dust intrusions below 1.5 km in winter (Jing et al., 2024). However, anthropogenic aerosols in winter are usually concentrated below 1.5 km (Jing et al., 2025). Therefore, the winter analyzed here represent hygroscopic growth effect at higher altitudes in the free troposphere, where the atmosphere is relatively clean, rather than surface-level pollution. Chen et al. (2019) reported an extremely low $\gamma$ of 0.1 under clean conditions, revealing that the hygroscopicity of clean air is lower than that of polluted air. Similarly, Sicard et al. (2022) found only slight seasonal variation

in $\gamma$ values in Barcelona, with a maximum of 0.58 in summer and a minimum of 0.53 in autumn. They interpreted this season-independent $\gamma$ as recirculation layers of pollutants above the BL, caused by strong insolation, weak synoptic forcing, sea breezes, and mountain-induced winds (Pérez et al., 2004).

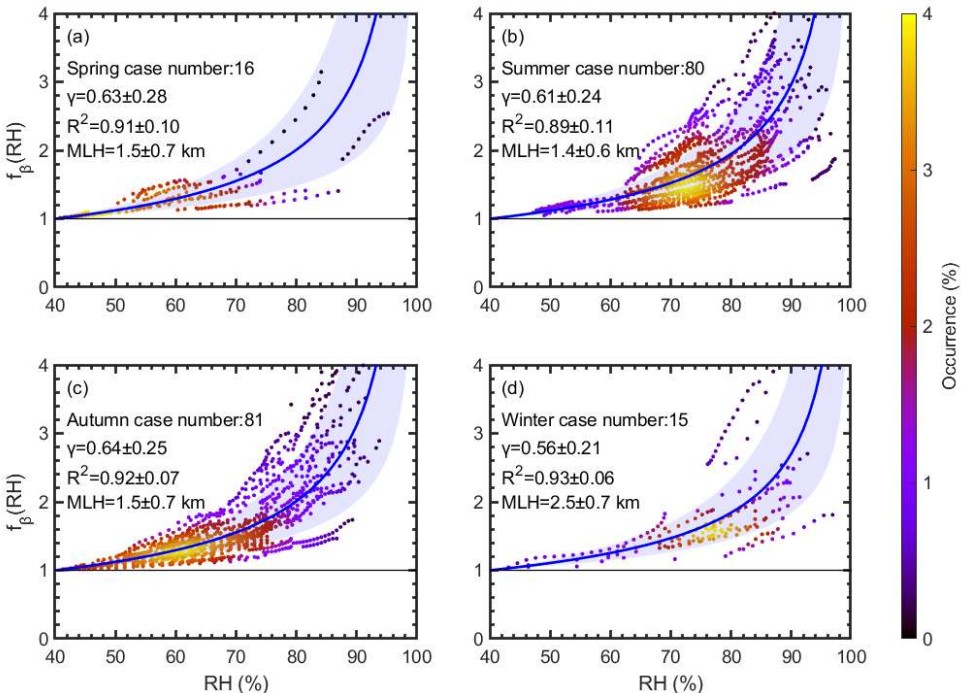

**Figure 6. Probability density distribution of the particle backscatter coefficient enhancement factors (RH$_{ref}$=40%) of selected**
**hygroscopic growth cases in (a) spring, (b) summer, (c) autumn, and (d) winter. The Hänel fits (blue solid curve) were calculated with the mean hygroscopic growth parameter $\gamma$ for each season. The shaded areas represent the standard deviation of Hänel fits.**

Table 3 summarizes hygroscopic growth parameters from this study in Wuhan and from the existing literature measured elsewhere. A variety of aerosol mixtures was examined, with $\gamma$ values varying from 0.24 for mixtures dominated by hydrophobic particles, such as dust, to 1.1 for highly hygroscopic aerosols (e.g., marine aerosols and inorganic salts). To
analyze the causes of the lidar-observed aerosol hygroscopic growth, many studies indirectly inferred the aerosol composition through backward trajectories and optical properties (e.g. Ångström exponent, complex refractive index, depolarization ratio) (Veselovskii et al., 2009; Granados-Muñoz et al., 2015; Sicard et al., 2022; Haarig et al., 2025). Miri et al. (2024) introduced fluorescence capacity, which was not affected by water vapor, to distinguish aerosol components (with biological aerosols exhibiting higher fluorescence, and pure dust or urban aerosols demonstrating lower fluorescence). Ground-based aerosol
chemical speciation monitor (ACSM) was also used to explain the hygroscopic growth behavior of aerosols through ground-level chemical composition analysis (Lv et al., 2017; Bedoya-Velásquez et al., 2018; Chen et al., 2019; Wu et al., 2020). Pérez-Ramirez et al. (2021) provided the first airborne in situ measurements for chemical composition determination, confirming that sulfates and water-soluble organic carbon are the main contributors to the aerosol hygroscopic growth observed by lidar,

with $\gamma$ values of 0.38-0.39. In addition, Laly et al. (2025) combined lidar-based hygroscopic growth estimation with chemical
species data from the Copernicus atmospheric monitoring service (CAMS), showing that $\gamma$ was significantly higher in regional pollution cases affected by sea salt (0.87 and 1.52) compared with those without sea salt influence (0.30-0.75).

The $\gamma$ value from this study (0.62±0.24) is slightly higher than that reported by Sicard et al. (2022) (0.55±0.23), who conducted a statistical analysis of hygroscopic growth parameters for local/regional pollutants and sea salt aerosols without dust interference in Barcelona, Spain. Several factors may explain this difference. First, although sea salt was not considered
in the present study, we speculate that the diverse inorganic salts emitted from extensive human activities in Wuhan probably contributed significantly to the observed strong hygroscopicity. Liu et al. (2014) found that the fractions of ammonium, nitrates, and sulfate are strongly correlated with aerosol hygroscopicity. Similarly, He et al. (2016) identified a region of high hygroscopic growth in Eastern China, corresponding to large-scale industrial districts with substantial emissions of inorganic salts, such as sulfates and nitrates. Wu et al. (2020) also measured a $\gamma$ value of 1.14 for fine-mode inorganic salts in urban
pollution over Beijing. Second, both Sicard et al. (2022) and this study are based on lidar observations within two hours before and after the radiosonde launches (00 and 12 UTC). During this period, Barcelona corresponds to noon or midnight (UTC+1 in winter and UTC+2 during daylight saving time). In contrast, Wuhan (UTC+8) experienced the morning and evening rush hours, which contribute substantially to traffic emitted $NO_2$. Zhang and Cao (2015b) found two $NO_2$ emission peaks in Chinese megacities (Beijing, Shanghai, and Guangzhou, which have traffic patterns similar to Wuhan) between 7-10 and 19-22 local
time. As discussed previously, higher $NO_2$ emissions favor the formation of hygroscopic nitrate particles, contributing to the larger $\gamma$ values observed in Wuhan.

**Table 3. Comparisons of the 532-nm lidar-estimated hygroscopic growth parameter $\gamma$ at $RH_{ref}$=40%, obtained using the Hänel fitting method for pollutants or aerosol mixtures. The results from this study alongside those reported in previous studies are provided.**

| Research type | Location | Instrument | Aerosol mixture | $\gamma$ | Reference |
|---|---|---|---|---|---|
| Statistics (192 cases) | Wuhan, China (30.5 °N, 114.4°E) | polarization lidar | Local/regional pollution | 0.62±0.24 | This study |
| Statistics (76 cases) | Barcelona, Spain (41.2 °N, 2.1 °E) | multi-wavelength lidar micro-pulse lidar | Local pollution, sea salt Local/regional pollution, sea salt | 0.55±0.23 | Sicard et al. (2022) |
| Case study | Granada, Spain (37.2 °N, 3.6 °E) | multi-wavelength Raman lidar | Marine aerosols, sulfates | 1.10 | Granados-Muñoz et al. (2015) |
| | | | Marine aerosols, sulfates, smoke, dust | 0.56 | |
| Case study | Granada, Spain (37.2 °N, 3.6 °E) | multi-wavelength Raman lidar | Smoke, urban pollution | 0.48 | Bedoya-Velásquez et al. (2018) |
| Case study | Baltimore–Washington DC, USA (38.99°N, 76.84°W) | multiwavelength Mie–Raman lidar | Sulfate | 0.9 | Veselovskii et al. (2009) |

| Case study | Baltimore–Washington DC, USA (38.99°N, 76.84°W) | multiwavelength Mie–Raman lidar | Sulfate, water-vapor-soluble organic carbon (more organic content) | 0.39 | Pérez-Ramírez et al. (2021) |
| | | | Sulfate, water-vapor-soluble organic carbon | 0.38 | |
| Case study | Xingtai, China (37 °N, 114 °E) | Raman lidar | Organics, nitrates, sulfates | 0.65 | Chen et al. (2019) |
| Case study | | | Clean condition | 0.10 | |
| Case study | Xinzhou, China (38.4 °N, 112.7 °E) | three-wavelength Mie polarization Raman lidar | Dust, organic, inorganic salts | 0.24 | Lv et al. (2017) |
| | | | Anthropogenic aerosol, organic, inorganic salts | 1.09 | |
| Case study | Beijing, China (39.5 °N, 116.2 °E) | micro-pulse lidar Raman lidar | Dust, organic, inorganic salts | 0.30 | Wu et al. (2020) |
| | | | Organic, inorganic salts | 1.14 | |
| Case study | Leipzig, Germany (51.3 °N, 12.3 °E) | Raman–polarization lidar | Continental aerosol | 0.45 | Haarig et al. (2025) |
| Case study | Cabauw, Netherlands (52.0 °N, 4.9 °E) | multi-wavelength Raman lidar | Organics, nitrates, marine aerosols | 0.88 | Fernández et al. (2015) |
| | | | Organics, nitrates | 0.59 | |
| Case study | Lille, France (50.6 °N, 3.1 °E) | Mie–Raman–fluorescence lidar | Urban pollution | 0.47 | Miri et al. (2024) |
| | | | Smoke | 0.50 | |
| Case study | Saclay, France (48.7 °N, 2.1 °E) | Water Vapour and Aerosols Lidar | Regional pollution | 0.30-0.75 | Laly et al. (2025) |
| Case study | Paris, France (48.8 °N, 2.3 °E) | | Sea salts, regional pollution | 0.87, 1.52 | |

## 4.2 The uncertainty introduced by assuming a fixed lidar ratio

In Section 2, the backscatter coefficient $\beta_p$ is retrieved by the Fernald method with a fixed lidar ratio (LR) of 50 sr (Fernald, 1984). However, the hygroscopic growth process can lead to an increase in LR under high RH conditions in previous studies: Veselovskii et al. (2025) found that during the hygroscopic growth, the extinction increases more rapid than the backscatter; Haarig et al. (2025) estimated the LR enhancement factor of 1.43 when RH=90%. While the polarization lidar cannot measure the LR for specific cases like the Raman lidar. Zhao et al. (2017) have estimated a relational expression between LR and RH through 532-nm micro-pulsed lidar and Mie model:

$$\text{LR} = \text{LR}_{dry} \times (0.92 + 2.5 \times 10^{-2}(\text{RH} - 40) - 1.3 \times 10^{-3}(\text{RH} - 40)^2 + 2.2 \times 10^{-5}(\text{RH} - 40)^3) \tag{7}$$

where $\text{LR}_{dry}$ represents the LR under dry conditions. The fixed LR of 50 sr represents an average value derived from combined lidar and sun photometer measurements in the ambient troposphere (Takamura et al., 1994). Based on radiosonde data, the average RH in the lower troposphere over Wuhan is approximately 40~70% (Guo et al., 2023). Accordingly,

$LR_{dry}=47$ sr in Eq. (7) was setted, such that an LR of 50 sr corresponds to RH values of approximately 50~55%. It should be mentioned that our analysis focus on the variation of LR within the identified particle-hygroscopic-growth layers. Taking the case from July 19, 2019, presented in Section 3 as an example, figure 7 illustrates the influence of variable LR on hygroscopic parameter $\gamma$. The LR increases with rising RH, and the derived backscatter coefficient (in red curve) shows a slight deviation from the original profile (in black curve). The derived $\gamma$ increases by 2.1% from 0.48 to 0.49. Furthermore, Table 4 summarizes 10 cases covering RH ranges of 40%-100%. The variable LR generally causes an increase in $\gamma$ of <10%, with the largest increase up to 12.5% under high RH conditions. The uncertainty introduced by assuming a fixed LR becomes more prouounced under higher RH conditions.

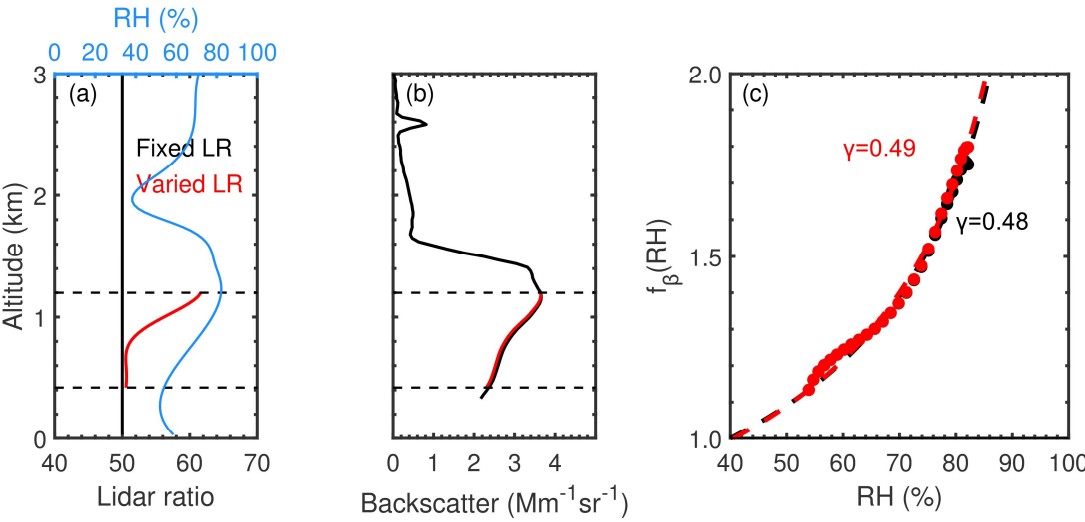

**Figure 7.** A case to illustrate the difference of hygroscpic parameter $\gamma$ between using a fixed LR and a variable LR over Wuhan at 1830-1900 LT on 19 July 2019. Profiles of (a) lidar ratio and RH, (b) backscatter coefficient; (c) the particle backscatter coefficient enhancement factors calculated by the Hänel method are presented. The black and red lines represent profiles derived by a fixed LR of 50 sr and variable LR, respectively.

**Table 4. Comparisons of hygroscopic growth parameter $\gamma$: fixed LR versus variable LR.**

| Date | RH range | $\gamma_1$ by fixed LR | $\gamma_2$ by variable LR | $\frac{\gamma_2-\gamma_1}{\gamma_1}$ |
|------|----------|------------------------|---------------------------|--------------------------------------|
| 2013.07.09 | 71~84 % | 0.42 | 0.46 | 9.5 % |
| 2018.08.21 | 84~95 % | 0.48 | 0.54 | 12.5 % |
| 2018.10.17 | 65~80 % | 0.53 | 0.56 | 5.7 % |
| 2019.07.19 | 54~82 % | 0.48 | 0.49 | 2.1 % |
| 2019.08.09 | 72~96 % | 0.45 | 0.50 | 11.1 % |
| 2020.08.15 | 72~89 % | 0.37 | 0.41 | 10.8 % |
| 2021.10.03 | 42~58 % | 0.70 | 0.72 | 2.9 % |
| 2022.10.01 | 68~89 % | 0.63 | 0.66 | 4.8% |
| 2022.11.10 | 53~68 % | 0.22 | 0.22 | 0 % |
| 2024.01.04 | 59~77 % | 0.49 | 0.53 | 8.2 % |

## 5 Summary and conclusions

In this study, we analyzed the statistical characteristics of the hygroscopic growth parameter $\gamma$ over Wuhan during 2010-2024. The dataset is based on 532-nm ground-based polarization lidar observations, meteorological data from radiosonde measurements, and ERA5 reanalysis. Simultaneous lidar-derived particle backscatter coefficients and radiosonde RH profiles were matched, and the use of meteorological parameters allowed the application of stringent constraints to the dataset. This approach identified 192 suitable cases for our analysis. The Hänel parameterization method was employed to estimate the hygroscopic growth parameter $\gamma$. A representative case observed on 19 July 2019 is presented to illustrate the methodology for identifying hygroscopic growth cases and estimating the hygroscopic growth parameter $\gamma$. In this case, $\gamma$ was 0.48, suggesting moderately hygroscopic particles, typical of urban pollution. The corresponding $f_{ref\ änel}(85\%)$ value was 1.93, showing that the backscatter coefficient increases by a factor of 1.93 as RH rose from 40% (dry condition) to 85%.

For the statistical characteristics, the average and standard deviations of $\gamma$ were 0.62±0.24, corresponding to an $f_{ref-\ änel}(85\%)$ value of 2.36, with a range of 1.69-3.29 when incorporating the standard deviation. All identified cases were classified by altitudes into the boundary layer and free troposphere clusters. No significant difference in $\gamma$ was observed between the BL (0.63±0.25) and FT (0.60±0.24). The hygroscopic growth of anthropogenic aerosols in Wuhan generally occurred under high RH conditions around 60-80%. The annual mean $\gamma$ increased sharply from 0.49 in 2014 to 0.63 in 2017 and stabilized between 0.6 and 0.7 after 2018. This trend closely matches the evolution of the annual mean $NO_2$-to-$SO_2$ concentration ratio, which rose from 1.8 in 2014 to 5.3 in 2017 and was situated between 4 and 6 after 2018. These results indicate that the presence of nitrates in the aerosol mixture enhanced hygroscopicity under similar RH conditions (Chen et al., 2019). Regarding seasonal variation, most cases occurred in summer (80) and autumn (81). The seasonal average $\gamma$ showed minimal variation, with a minimum in winter (0.56) and a maximum in autumn (0.64), corresponding to $f_{ref-\ änel}(85\%)$ values of 2.17 and 2.43, respectively, i.e., a difference of approximately 11%. The lower $\gamma$ (0.56) in winter is due to the higher MLH of 2.4 km compared with other seasons, indicating that the winter analyzed cases reflect hygroscopic growth effect of relatively clean aerosols at higher altitudes rather than severe surface-level pollution. Finally, we tried to estimate the error introduced by a fixed LR. The incorporation of variable LR generally causes an increase in $\gamma$ of <10%, with the largest increase reaching up to 12.5% under high RH conditions.

Leveraging long-term polarization lidar observations, we characterize the hygroscopicity of local/regional pollutants over Wuhan. The hygroscopic growth parameter in this study demonstrates how RH amplifies lidar-derived backscatter coefficients and implies the potential influence on long-term AOD variation. In our previous work (Jing et al., 2025), long-term lidar observations over Wuhan from 2010 to 2024 revealed a two-stage evolution of anthropogenic aerosols: a rapid decline from 2010 to 2017, followed by a fluctuating period during 2018-2024. The larger $\gamma$ values after 2017 may have amplified the hygroscopic growth effect on the backscatter coefficient, and thus also the integrated AOD, which may partially offset the efforts of emission control policies and contribute to the cessation of the AOD decline post-2018. Furthermore, AOD is a major source of uncertainty in estimates of direct aerosol radiative forcing (DARF) (Elsey et al., 2024); in future work, we will

assess the influence of particle hygroscopic growth on DARF. However, polarization lidar cannot distinguish aerosol chemical composition in urban environments. To better assess the hygroscopic parameter $\gamma$ for specific aerosol types, it would be beneficial to combine HYSPLIT trajectory simulations, chemical analysis instruments, satellite data, and Raman lidar observations. In addition, radiosondes are launched approximately 24 km away from our lidar site at 08 LT and 20 LT, limiting the availability of co-located and simultaneous meteorological data (He et al., 2023). Raman lidar can provide real-time and co-located, height-resolved measurements of temperature and RH (Liu et al., 2019; Dawson et al., 2020; Pan et al., 2020; Yi et al., 2021), enabling a more accurate analysis of aerosol hygroscopicity property over Wuhan.

**Data availability**

ERA5 reanalysis data can be obtained from https://cds.climate.copernicus.eu/datasets (Copernicus Climate Change Service, Climate Data Store, 2025). The radiosonde data can be obtained from https://weather.uwyo.edu/upperair/sounding.shtml (University of Wyoming Atmospheric Science Radiosonde Archive, 2025). The HYSPLIT model is available at https://www.arl.noaa.gov (HYSPLIT, 2025). Lidar data used to generate the results of this paper are available from the authors upon request (e-mail: yf@whu.edu.cn).

**Author contributions**

YH, DJ, and ZY analyzed the data and wrote the manuscript. ZY, DM, and KH participated in scientific discussions and reviewed and proofread the manuscript. YH and FY conceived the research and acquired the research funding. FY led the study.

**Competing interests**

The contact author has declared that none of the authors has any competing interests.

**Financial support**

This work was supported by the National Natural Science Foundation of China (grant nos. 42575138, 42005101, 42575141, and 42205130), Hubei Provincial Special Project for Central Government Guidance on Local Science and Technology Development (2025CFC003), and the Meridian Space Weather Monitoring Project (China).

**Acknowledgements**

The authors thank the colleagues who participated in the operation of the lidar system at our site. We also acknowledge the European Centre for Medium-Range Weather Forecasts (ECMWF) for ERA5 reanalysis data, the University of Wyoming for radiosonde data, and the National Oceanic and Atmospheric Administration (NOAA) Air Resources Laboratory (ARL) for the
HYSPLIT model.

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
