# Peer review of "Hygroscopic growth characteristics of anthropogenic aerosols over central China revealed by lidar observations"

_EGUsphere, 2025_

## Author Response (AR1)

**Responses to RC1**

**General Remarks**

This manuscript presents a valuable long-term study of aerosol hygroscopicity in Wuhan, Central China, utilizing a 15-year dataset of ground-based polarization lidar observations. The authors successfully identify 192 cases of hygroscopic growth and provide a detailed statistical analysis of the hygroscopic growth parameter, exploring its inter-annual trends, seasonal variations, and vertical distribution. The contribution of this work is significant. Long-term, vertically resolved datasets of aerosol hygroscopicity are rare, and this study offers critical insights into how emission control policies, specifically the shifting ratio of $NO_2$ to $SO_2$, may be altering the optical properties of urban aerosols over time. The approach of combining lidar with radiosonde and reanalysis data is sound, and the manuscript is generally well-structured and clear. However, to ensure the robustness of the retrieved optical properties and the subsequent hygroscopic parameters, I have a few specific concerns regarding the retrieval assumptions and aerosol classification methods. Addressing the following points would strengthen the physical interpretation of the results. I hope my comments can be helpful in refining this interesting study.

**Response:** We appreciate your thoughtful review and valuable comments on our manuscript. We have added the specific depolarization ratio used to separate dust and non-dust components and discussed the potential interference from internally mixed dust. In addition, a sensitivity analysis have been conducted to assess how varibility in lidar ratio affects the derived hygroscopic growth parameter. Point-by-point responses are provided below, and the manuscript has been revised accordingly.

**Specific comments**

**Comment:** The study utilizes the POLIPHON method to separate dust and non-dust (anthropogenic) components. However, the manuscript does not explicitly state the specific particle depolarization ratios for pure dust and non-dust used for this separation. Since the derived non-dust backscatter coefficient is highly sensitive to these threshold values, please explicitly list them in the methodology section.

**Response:** Thank you very much for your suggestion. We have added statements specifying the particle depolarization ratios used to separate dust and non-dust compnents. The following sentences have been added in the revised manuscript "**In addition, the non-dust particle backscatter coefficient $\beta_{nd}$ is calculated using the polarization-lidar photometer networking (POLIPHON) method as follows (Tesche et al., 2009; Mamouri and Ansmann, 2014):**

$$\beta_{nd}(z) = \beta_p(z) - \beta_p(z) \frac{[\delta_p(z) - \delta_{nd}](1 + \delta_d)}{(\delta_d - \delta_{nd})[1 + \delta_p(z)]} \tag{1}$$

**where z represents the altitude; $\delta_d = 0.31$ and $\delta_{nd} = 0.05$ are the particle depolarization ratios for dust and non-dust, respectively. For each profile, we set $\beta_{nd}(z) = \beta_p(z)$ if $\delta_p(z) < \delta_{nd}$ and $\beta_{nd}(z) = 0$ if $\delta_p(z) > \delta_d$ (Mamouri and Ansmann, 2014).**" (please see lines 111-117) The values of $\delta_d = 0.31$ and $\delta_{nd} = 0.05$ are summarized from previous lidar observations in East Asia (Sugimoto et al., 2003; Shimizu et al., 2004). Moreover, laboratory studies by Sakai et al. (2010) estimated the particles depolarization ratio of 0.39 for coarse-mode particles and 0.17 for fine-mode particles for Asian dust, which are consistent with observational results when assuming a 25% contribution from fine-mode dust.

**Comment:** The methodology assumes that dust and non-dust aerosols are externally mixed. In a humid environment like Wuhan, it is common for dust particles to become internally mixed (e.g., coated) with soluble pollutants during transport. This coating process typically lowers the particle depolarization ratio of the dust. Could the POLIPHON method be misclassifying these "coated dust" particles as "anthropogenic" due to their

reduced depolarization? If misclassification occurs, the "non-dust" component would be contaminated by less hygroscopic dust cores. Could this artificially lower the reported hygroscopicity of the anthropogenic component? A brief discussion on this potential contamination and its impact on the final statistics would be very helpful.

**Response:** Thank you very much for pointing this out. You are correct that the POLIPHON method does not account for internal mixing situations, which may lead to the misclassification of "coated dust" particles as "anthropogenic". However, this potential misclassification does not significantly affect our results for the following reasons.

First, Fig. 1R(a) shows the dust intrusion conditions of the 192 analyzed cases. Approximately 54.7% of the cases show no dust interference, and for the remaining cases the dust optical depth (DOD) does not exceed 0.05. This is consistent with our findings that most hygroscopic growth cases are identified in summer and autumn (accounting for 83.5% of all cases), when Wuhan is least affected by dust intrusions and typically exhibits low DOD values (Jing et al., 2024).

Second, to estimate the potential influence of internally mixed dust, we conducted a sensitivity analysis. Given that the $\delta_p$ during dust events over Wuhan varies between 0.1 and 0.3 (Jing et al., 2024), we reduced the threshold value for pure dust $\delta_d$ (as defined in the response above) to lower values of 0.10~0.25, thereby making the extraction of "anthropogenic aerosols" more conservative. Figure 1R(b) shows the derived hygroscopic parameter $\gamma$ using different $\delta_d$ valuese. Even when $\delta_d$ is reduced to 0.10, $\gamma$ for all cases changes only marginally, increasing by 0.02 (0.62 to 0.64).

In summary, although the misclassification of internally mixed dust could slightly lower the estimated hygroscopicity of the anthropogenic component, it does not have a significant impact on the conclusions in the study. The corresponding discussions have been added to the revised manuscript. (please see lines 248-255)

[Figure]

**Figure 1R. (a)Histogram of DOD for the 192 analyzed cases; (b) hygroscopic parameter $\gamma$ at different threshold $\delta_d$ values.**

**Comment:** The particle backscatter coefficient is retrieved using a fixed lidar ratio of 50 sr. This assumption requires further elaboration. As aerosols absorb water and grow, their microphysical properties (size distribution, refractive index, shape) change, which typically causes the lidar ratio to vary rather than remain constant. Please discuss or quantify the potential error introduced by holding this value fixed during humidification events. A simple sensitivity analysis showing how a varying lidar ratio affects the calculated hygroscopic growth parameter would improve the robustness of the conclusions.

**Response:** In light of the reviewer's comments, we have added a sensitivity analysis as follows. Since polarization lidar cannot directly retrieve the lidar ratio (LR), we adopt the LR-RH (relative humidity) relationship reported by Zhao et al. (2017):

$$\text{LR} = \text{LR}_{dry} \times (0.92 + 2.5 \times 10^{-2}(\text{RH} - 40) - 1.3 \times 10^{-3}(\text{RH} - 40)^2 + 2.2 \times 10^{-5}(\text{RH} - 40)^3) \quad (2)$$

where $LR_{dry}$ represents the LR under dry conditions. The fixed LR of 50 sr represents an average value derived from combined lidar and sun photometer measurements in the ambient troposphere (Takamura et al., 1994). Considering that the average RH in the lower troposphere over Wuhan is approximately 40~70% based on radiosonde measurements (Guo et al., 2023), we set $LR_{dry}$=47 sr in Eq. (2), such that an LR of 50 sr corresponds to RH values of approximately 50~55%. It should be mentioned that our analysis focuses on the variation of LR within the identified particle-hygroscopic-growth layers. Taking the case presented in Section 3 as an example, LR varies with RH within the identified layer (between the two horizontal dashed lines in Figure 2R(a)). The revised backscatter coefficient (red curve) shows only a slight deviation from the original values obtained using a fixed LR (black curve in Figure 2R(b)). As a result, the hygroscopic growth $\gamma$ increases by 2.1% from 0.48 to 0.49 (Figure 2R(c)).

Furthermore, Table 1R summarizes 10 cases spanning RH ranges from 40% to 100%. Overall, $\gamma$ increases by up to 12.5% when using a variable LR instead of a fixed value. The uncertainty introduced by assuming a fixed LR becomes more prouounced under higher RH conditions. This sensitivity analysis and the corresponding discussions have also been added to the revised manuscript. (please see lines 330-354)

[Figure]

**Figure 2R. A case to illustrate the difference of hygroscpic parameter $\gamma$ between using a fixed LR and a variable LR over Wuhan at 1830-1900 LT on 19 July 2019. Profiles of (a) lidar ratio and RH, (b) backscatter coefficient; (c) the particle backscatter coefficient enhancement factors calculated by the Hänel method are presented. The black and red lines represent profiles derived by a fixed LR of 50 sr and variable LR, respectively.**

**Table 1R. Comparisons of hygroscopic growth parameter $\gamma$: fixed LR versus variable LR.**

| Date | RH range | $\gamma_1$ by fixed LR | $\gamma_2$ by variable LR | $\frac{\gamma_2 - \gamma_1}{\gamma_1}$ |
|------|----------|------------------------|---------------------------|----------------------------------------|
| 2013.07.09 | 71~84 % | 0.42 | 0.46 | 9.5 % |
| 2018.08.21 | 84~95 % | 0.48 | 0.54 | 12.5 % |
| 2018.10.17 | 65~80 % | 0.53 | 0.56 | 5.7 % |
| 2019.07.19 | 54~82 % | 0.48 | 0.49 | 2.1 % |
| 2019.08.09 | 72~96 % | 0.45 | 0.50 | 11.1 % |
| 2020.08.15 | 72~89 % | 0.37 | 0.41 | 10.8 % |
| 2021.10.03 | 42~58 % | 0.70 | 0.72 | 2.9 % |
| 2022.10.01 | 68~89 % | 0.63 | 0.66 | 4.8% |
| 2022.11.10 | 53~68 % | 0.22 | 0.22 | 0 % |
| 2024.01.04 | 59~77 % | 0.49 | 0.53 | 8.2 % |

**References:**

Guo, X., Huang, K., Fang, J., Zhang, Z., Cao, R., and Yi, F.: Seasonal and Diurnal Changes of Air Temperature and Water Vapor Observed with a Microwave Radiometer in Wuhan, China, Remote Sens., 15, 5422, https://doi.org/10.3390/rs15225422, 2023.

Jing, D., He, Y., Yin, Z., Liu, F., and Yi, F.: Long-term characteristics of dust aerosols over central China from 2010 to 2020 observed with polarization lidar. Atmos. Res., 297, 107129, https://doi.org/10.1016/j.atmosres.2023.107129, 2024.

Mamouri, R. E. and Ansmann, A.: Fine and coarse dust separation with polarization lidar, Atmos. Meas. Tech., 7, 3717–3735, https://doi.org/10.5194/amt-7-3717-2014, 2014.

Sakai, T., Nagai, T., Zaizen, Y., and Mano, Y.: Backscattering linear depolarization ratio measurements of mineral, sea-salt, and ammonium sulfate particles simulated in a laboratory chamber, Appl. Optics, 49, 4441–4449, https://doi.org/10.1364/AO.49.004441, 2010.

Shimizu, A., Sugimoto, N., Matsui, I., Arao, K., Uno, I., Murayama, T., Kagawa, N., Aoki, K., Uchiyama, A., and Yamazaki, A.: Continuous observations of Asian dust and other aerosols by polarization lidars in China and Japan during ACE-Asia, J. Geophys. Res., 109, D19S17, https://doi.org/10.1029/2002JD003253, 2004.

Sugimoto, N., Uno, I., Nishikawa, M., Shimizu, A., Matsui, I., Dong, X., Chen, Y., and Quan, H.: Record heavy Asian dust in Beijing in 2002: Observations and model analysis of recent events, Geophys. Res. Lett., 30, 1640, https://doi.org/10.1029/2002GL016349, 2003.

Takamura, T., Sasano, Y., and Hayasaka, T.: Tropospheric aerosol optical properties derived from lidar, sun photometer, and optical particle counter measurements, Appl. Opt., 33, 7132–7140, https://doi.org/10.1364/AO.33.007132, 1994.

Tesche, M., Ansmann, A., Müller, D., Althausen, D., Engelmann, R., Freudenthaler, V., and Groß, S.: Vertically resolved separation of dust and smoke over Cape Verde using multiwavelength Raman and polarization lidars during Saharan mineral dust experiment 2008, J. Geophys. Res., 114, D13202, https://doi.org/10.1029/2009JD011862, 2009.

Zhao, G., Zhao, C., Kuang, Y., Tao, J., Tan, W., Bian, Y., Li, J., and Li, C.: Impact of aerosol hygroscopic growth on retrieving aerosol extinction coefficient profiles from elastic-backscatter lidar signals, Atmos. Chem. Phys., 17, 12133–12143, https://doi.org/10.5194/acp-17-12133-2017, 2017.

**Responses to RC2**

**General Remarks**

This work presents an excellent study about aerosol hygroscopic growth with lidar measurements. Its uniqueness is that it presents a long-term analysis of hygroscopic growth parameters with is very rare in the literature. In general, the paper is very well-structured and written.

**Response:** We appreciate the reviewer's thorough review and constructive comments. All comments have been carefully addressed in the revised manuscript. The corresponding modifications are highlighted in red, and our detailed responses to each comment are given below in blue.

**Specific comments**

**Comment:** I agree with all comments of previous referee, and I just want to add one question: If the authors claim they can infer the role of aerosol hygroscopicity in aerosol optical depth (i.e. conclusion section in lines 340-341), why not estimating its role in direct radiative forcing? They have all the required measurements to do so.

**Response:** We have responded to RC1's comments and revised the manuscript accordingly. Please refer to our response regarding RC1 for details. According to the reviewer's comment, the statements of direct radiative forcing have been added as "**Furthermore, AOD is a major source of uncertainty in estimates of direct aerosol radiative forcing (DARF) (Elsey et al., 2024); in future work, we will assess the influence of particle hygroscopic growth on DARF.**" (please see lines 386-388)

**Comment:** Section 2.2.- Polarization lidar and data processing: I acknowledge that the most relevant information about the system is already available in the literature. However, I miss here the critical information about the system (i.e. laser power, signal-to-noise ratio, FOV, detectors). I encourage you to improve this section

**Response:** In light of the reviewer's comments, the following statements have been added " **The lowermost height with complete field-of-view (FOV) observation is 0.3 km. Specifications of the polarization lidar system are listed in Table 1.**

**Table 1. Specifications of the polarization lidar system at Wuhan University. (He et al., 2024).**

| Transmitter | | Receiver | |
|---|---|---|---|
| **Laser** | **Continuum Inlite II-20** | **Telescope** | **300 mm Cassegrain** |
| **Wavelength** | **532 nm** | **Diameter** | **300 mm** |
| **Energy/pulse** | **~ 120 mJ** | **Field of view** | **1 mrad** |
| **Repetition rate** | **20 Hz** | **PMT** | **Hamamatsu 5783P** |
| **Pulse duration** | **6 ns** | **Digitizer** | **Licel TR40-160** |

"(please see lines 105-108)

**Comment:** Line 148: If I understand well, you use ERA-5 to determine BL and FT. I am right? If so, can you give some description of your methodology?

**Response:** ERA5 provides the boundary layer height (BLH, unit: m) directly (Hersbach et al., 2023). The relevant description of our methodology is presented in the manuscript *"Considering the presence of an aerosol residual layer, the diurnal maximum BLH from ERA5 was adopted as the boundary between the BL and FT."* (please see lines 155-156)

**Comment:** Line 160: I wonder if two hours interval is too large for the matchups

**Response:** We applied a two-hour interval so as to maintain consistency with the previous study by Sicard et al. (2022), and the corresponding citation has been added. (please see line 167)

**Comment:** Lines 168-169: Wind data are from ERA-5? Please specify

**Response:** Wind data are from radiosonde. Please see the relevant statements in the manuscript "*…the maximum variations in radiosonde meteorological parameters within the identified aerosol layer were constrained as follows to ensure analysis under well-mixed atmospheric conditions…*" (please see lines 170-172)

**Comment:** Line 203: I miss a reference for the claim that these values are typical of urban pollution.

**Response:** We have been added a reference of Bedoya-Velázquez et al. (2018). (please see line 209)

**Comment:** Line 202-204: It is not clear to me the claim about the role of marine aerosols. You certainly are too far from coastal sites.

**Response:** Thank you for pointing out this. We have claimed that the role of marine aerosols can be ignored, relevant statement is presented in the manuscript "*…marine aerosols (e.g. sea salt) have been largely removed by sedimentation, leaving an extremely limited influence of sea salt in Wuhan*" (please see lines 210-211) To avoid ambiguity, a statement about excluding marine aerosol in our manuscript is added in the ending of Section 3 "**In this study, only anthropogenic aerosols were considered, while the influence of natural aerosols, such as mineral dust or sea salt, was excluded. Hygroscopic growth parameters for mineral dust are known to be very low, with $\gamma$ values of 0.20 at 355 nm and 0.12 at 1064 nm (Navas-Guzmán et al., 2019). For marine aerosols, $\gamma$ has been estimated as 1.49 for pure sea salt (Haarig et al., 2017) and 1.1 for a mixture of sulfate and sea salt (Granados-Muñoz et al., 2015). As Wuhan is an inland city far from the ocean, the impact of marine aerosols is minimal; therefore, cases with $\gamma$ <0.2 or $\gamma$ >1.1 were treated as outliers and excluded from the analysis.** (please see lines 227-232)

**Comment:** Line 249: I miss a reference after the statement 'our previous study'

**Response:** A citation of Jing et al. (2025) has been added. (please see line 265)

**Comment:** Line 277-278: Again, to me this is confusing. You are too far from coastal area.

**Response:** These statements are the discussion of the results by Sicard et al. (2022). To avoid misleading readers, relevant statements are removed.
* * *
**References:**

Bedoya-Velásquez, A. E., Navas-Guzmán, F., Granados-Muñoz, M. J., Titos, G., Román, R., Casquero-Vera, J. A., Ortiz-Amezcua, P., Benavent-Oltra, J. A., de Arruda Moreira, G., Montilla-Rosero, E., Hoyos, C. D., Artiñano, B., Coz, E., Olmo-Reyes, F. J., Alados-Arboledas, L., and Guerrero-Rascado, J. L.: Hygroscopic growth study in the framework of EARLINET during the SLOPE I campaign: synergy of remote sensing and in situ instrumentation, Atmos. Chem. Phys., 18, 7001–7017, https://doi.org/10.5194/acp-18-7001-2018, 2018.

Elsey, J., Bellouin, N., and Ryder, C.: Sensitivity of global direct aerosol shortwave radiative forcing to uncertainties in aerosol optical properties, Atmos. Chem. Phys., 24, 4065–4081, https://doi.org/10.5194/acp-24-4065-2024, 2024.

He, Y., Jing, D., Yin, Z., Ohneiser, K., and Yi, F.: Long-term (2010–2021) lidar observations of stratospheric aerosols in Wuhan, China, Atmos. Chem. Phys., 24, 11431–11450, https://doi.org/10.5194/acp-24-11431-2024, 2024.

Hersbach, H., Bell, B., Berrisford, P., Biavati, G., Horányi, A., Muñoz Sabater, J., Nicolas, J., Peubey, C., Radu, R., Rozum, I., Schepers, D., Simmons, A., Soci, C., Dee, D., and Thépaut, J.-N.: ERA5 hourly data on single levels from 1940 to present, Copernicus Climate Change Service (C3S) Climate Data Store (CDS) [data set], https://doi.org/10.24381/cds.adbb2d47, 2023.

Sicard, M., Fortunato dos Santos Oliveira, D. C., Muñoz-Porcar, C., Gil-Díaz, C., Comerón, A., Rodríguez-Gómez, A., and Dios Otín, F.: Measurement report: Spectral and statistical analysis of aerosol hygroscopic growth from multi-wavelength lidar measurements in Barcelona, Spain, Atmos. Chem. Phys., 22, 7681–7697, https://doi.org/10.5194/acp-22-7681-2022, 2022.

---

## Author Response (AR2)

**Responses to RC1**

The authors have done an excellent job with this revision. They have thoroughly addressed all of my previous comments and concerns, and the manuscript is significantly strengthened. I recommend that the paper be accepted in its current form.

**Response:** Thank you very much for your previous valuable comments and recognition on our work.